# A Case of Paraphilia with Osteoporosis and Administered with Depot Leuprorelin

**DOI:** 10.3390/medicina55100705

**Published:** 2019-10-20

**Authors:** Dohee Kim, Kyoung Min Kim, Myung Ho Lim

**Affiliations:** 1Departments of Internal Medicine & Kinesiologic Medical Science, Dankook University, Cheonan 31116, Korea; dh9070@dankook.ac.kr; 2Department of Psychiatry, College of Medicine, Dankook University, Cheonan 31116, Korea; profuture@naver.com; 3Departments of Psychology & Psychotherapy, Environmental Health Center, College of Health Science, Dankook University, 119 Dandae-Rho, Dongnam-Gu, Cheonan 31116, Korea

**Keywords:** paraphilia, osteoporosis, hyperthyroidism, leuprorelin

## Abstract

Paraphilia is a complex psychological and psychiatric disorder that has been difficult to treat. Leuprorelin has been used as one of the therapeutic methods for paraphilia. Leuprorelin administration could change insulin resistance and accelerate bone loss. The case study in this work was a 59-year-old man who visited a hospital with the chief complaints of frotteuristic behaviors in public places, a continuous increase in sexual desire, and sexual molestation behavior that started in 2007. We injected leuprorelin (3.6 mg) intramuscularly every month for this patient with paraphilia and comorbidities of osteoporosis and hyperthyroidism. The clinical global impression (CGI), Sex Addiction Screening Test (SAST), Wilson Sex Fantasy Questionnaire (WSFQ), physical examination, and laboratory tests were performed. After 12 months of leuprorelin injection for paraphilia, we found a significant improvement in abnormal sexual behavior/desire without aggravation of osteoporosis/hyperthyroidism. Gonadotrophin-Releasing Hormone (GnRH) analogs could be used as alternative or supplementary treatment methods for paraphilia with osteoporosis/hyperthyroidism.

## 1. Introduction

According to the criterion of the Diagnostic and Statistical Manual of Mental Disorders, fifth edition (DSM-5), paraphilic disorder was described as “recurrent, intense sexually arousing fantasies, sexual urges, or behaviors generally involving nonhuman objects, the suffering or humiliation of oneself or one’s partner, or children or other nonconsenting persons that occur over a period of at least 6 months” [1]. In the DSM-5, paraphilic disorder was divided into eight categories (exhibitionistic disorder, fetishistic disorder, frotteuristic disorder, pedophilic disorder, transvestic disorder, voyeuristic disorder, sexual sadistic disorder, and sexual masochistic disorder). 

There were many treatment methods for paraphilia in sexual offenders. Such methods included cognitive behavior therapy, psychotherapy, surgical castration, and drug therapy to reduce sexual desire. However, there was a conflict between the opinion that punishment for sexual offenders is necessary for public safety, and that the most clinically effective treatment should be given, even if the subject is a criminal [2]. 

The Ministry of Justice of Korea enacted legislation regarding the pharmacological treatment for the sexual impulses of sexual offenders aged 19 or over to prevent recidivism in 2011 [3]. The legislation of 2011 only included sexual offenders who assaulted a child aged 16 or under; however, the legislation was revised to include all sexual offenders in 2013 [4]. 

Gonadotrophin Releasing Hormone analogs (GnRH) are artificial synthetic agents of natural GnRH decapeptides. This hormone is produced in the body in the hypothalamus and secreted into the hypophyseal gland directly. GnRH analogs induced a decrease in gonadotrophin cells, a decrease in progesterone secretion, and a mild decrease in follicle-stimulating hormone [5]. The representative commercial drugs of GnRH analogs were goserelin (3.75 mg, 11.25 mg), triptorelin (3.75 mg, 11.25 mg), and leuprorelin (3.75 mg, 11.25 mg). Among them, leuprorelin (leuprolide acetate), as one of the luteinizing hormone-releasing hormone analog depot drugs, has been used as a selective drug for central precocious puberty [6]. GnRH administration or androgen deprivation therapy (ADT) could increase fat mass, leading to insulin resistance and accelerate bone loss, causing increased fracture risk. ADT accelerated age-related decline in bone mineral density (BMD) and increased fracture risk [7]. Long-acting GnRH agents were advantageous for subjects with very poor compliance, or who refused drug administration. However, there were not enough studies regarding the safety and effects of long-acting GnRH depot agents compared to oral agents [7,8,9,10,11].

We injected leuprorelin, a GnRH depot drug, into a patient with an intractable paraphilic disorder and comorbid hyperthyroidism and osteoporosis for 12 months. We found significant decreases in sexual desire and unfavorable behavioral symptoms. To our knowledge, although there have been a few studies regarding the treatment outcomes of chemical castration on sex offenders with paraphilia in Korea [4,12], there has been no report of the treatment of a patient with paraphilia and comorbid hyperthyroidism and osteoporosis. 

## 2. Case Report

A 59-year-old man visited the clinic of Dankook University Hospital on 2 March 2016 with the chief complaints of frotteuristic behavior in a public place, a continuous increase in sexual desire, and sexual molestation involving compulsory physical contact that started in 2007. The patient had been diagnosed with paraphilic disorder, intermittent explosive disorder, and borderline intellectual function three years prior by a psychiatrist. The patient had received three years of treatment at the National Forensic Hospital due to a sexually violent crime and uncontrolled paraphilic behavior. In 2014, the patient was arrested by the police for forced sexual molestation by kissing a woman and touching her while sleeping in a sauna facility. The patient had often lied to their family since high school, often went away, and was unemployed. He reported that he had been prostituting twice a month for sexual desire relief. The treatment of the patient was approved by the Institutional Review Board of Dankook University Hospital. A psychiatrist met him face to face, administered a full verbal explanation and written document about the purpose and procedure of this study, and received informed written consent to participate voluntarily. According to the guidelines of the World Federation of Societies of Biological Psychiatry (WFSBP), we rated the patient as Level 5 because of his high risk of sexual assault crimes and a high degree of paraphilic traits after release. We evaluated the patient’s state more seriously than Stage 4, which is the general sex hormone oral administration stage, and administered the GnRH injection according to these guidelines.

The patient had been diagnosed with Graves’ disease one year prior and had been taking 75–100 mg/day of propylthiouracil (PTU). In addition, the patient had been diagnosed with osteoporosis 10 years prior and had been taking 70 mg of alendronate weekly and 600 mg calcium/400 IU vitamin D 400 twice a day. The patient had no other disease relevant to his internal medicine or neurology. The patient did not drink and had stopped smoking two years ago. At the physical examination, his height was 164 cm, his weight was 65 kg, and his body mass index was 24.17 kg/m^2^. In the cardiovascular examination and complete blood count/liver function test examination, no abnormal findings were observed. The values of his hormonal test and thyroid function test at baseline and 6 and 12 months were shown in Table 1. His Luteinizing hormone (LH) and follicle-stimulating hormone (FSH) concentrations were increased above the reference level. The level of serum c-telopeptide was within the normal range, which is thought to be suppressed by the administration of alendronate. The baseline thoracolumbar spine X-ray showed degenerative spondylosis, diffuse osteopenia without compressive fracture, and his bone mineral density (BMD) showed osteoporosis (Table 1).

For the treatment of Graves’ disease, we changed the daily administration of 100 mg PTU to a daily administration of 10 mg methimazole (MMI), which gradually decreased to a daily 2.5 mg according to the result of the thyroid function test (TFT). For the treatment of the patient’s osteoporosis, we educated the patient about appropriate calcium/vitamin D intake and exercise and prescribed a weekly administration of 70 mg alendronate and the daily administration of calcium/vitamin D 500 mg/1000 IU (Table 1). We performed alendronate, calcium/vitamin D treatment at the same time as the Leuprorelin depot administration. 

After the leuprorelin injection, he reported that his sexual desire was too low. He also reported that he was not interested in sex and that he did not watch any sexual videos, and sexual erections also decreased to 1–2 instances per month. To evaluate the effect of treatment, the patient was assessed with the Clinical Global Impression (CGI), Sex Addiction Screening Test (SAST), and Wilson Sex Fantasy Questionnaire (WSFQ) at baseline and six months and one year after the leuprorelin administration, respectively. The patient was administered with 3.75 mg of long-acting leuprorelin first on 8 March 2016 and every four weeks. In addition, supportive psychotherapy was also implemented for the patient every two weeks. The sexual interest/activity and CGI severity/improvement were evaluated with the sex offender at the baseline state, month 6, and month 12 after starting the Leuprolide injections. The CGI severity score was 5 at the baseline state, 2 in month 6, and 1 in month 12. The CGI improvement score was 1 at month 6, and 3 in month 12 (Table 2). The total score for the SAST was 13 in the baseline state, 2 in month 6, and 1 in month 12, and WSFQ was 13 in the baseline state, 2 in month 6, and 1 in month 12 (Table 2).

His testosterone level was below 0.25 ng/mL after the administration, and it remained at a low level at six months and one year. The weight and blood pressure were continuously within normal limits until one year after the administration of the injection. In the follow-up blood test, the hemoglobin level was 11.4–12.4 mg/dL, indicating mild anemia, and the glucose and lipid levels were found to be within normal limits. The values of bone marrow density (BMD) after 6 and 12 months compared to the baseline were −4.7% and 0% for L1–4, −1.9% and −0.4% for the femur neck, and −6.1% and +4.6% for the total femur, respectively (coefficients of variation 1.0%, the least significant change >2.77%). The serum CTx level increased to 0.084 ng/mL and 0.155 ng/mL after 6 and 12 months, respectively. The serum 25(OH) D level was 27.5 ng/mL after 6 months, but it decreased to 11.0 ng/mL after 12 months. This change was thought to be due to the seasonal effect of less sunshine. Then, we increased the dose of vitamin D to 2800 IU weekly. 

The patient has not experienced a fracture for more than one year after the administration of the injection, and the weekly administration of 70 mg alendronate and the calcium/vitamin D supplement have been continuously maintained. During the one-year period after the administration of the injection, the patient did not experience any side-effects of the drug (e.g., worsening of osteoporosis and liver function disorder). After the administration of the leuprorelin depot, the patient reported that he was able to have an erection (0–2 times/month), but ejaculation was impossible, possibly due to the effect of this drug. In addition, after drug administration, the patient exhibited a rather feminine appearance change, such as smooth skin and a soft facial contour. There was also an emotional irritation, which was observed in the early stage of the outpatient treatment, and the treatment compliance was also appropriately maintained.

### Ethics Statement

The patient provided written informed consent for inclusion before he participated in the study. The study was conducted in accordance with the Declaration of Helsinki, and the protocol was approved by the Ethics Committee of Dankook University Hospital (Project identification code: DKU 2015-12-014).

## 3. Discussion and Conclusions 

This was the first report of clinical treatment with Leuprolide for paraphilic disorder with comorbid hyperthyroidism/osteoporosis. Previous case studies [8] reported that when triptorelin was injected into six patients with paraphilia for seven years, every abnormal sexual behavior disappeared in five patients [10]. Also, triptorelin was administered to 30 patients with paraphilia for 8–42 months [9], and the authors observed a remarkable improvement in the abnormal sexual fantasy/desire scale score in these patients, from 48 points before treatment to 5 points after treatment. We chose the leuprorelin agent because it is the most commonly used injection drug for paraphilia in the context of sexual crimes. It is also highly reliable because of its European certification in the treatment of precocious puberty. In this case study, we found a 4-point improvement in the CGI scale and a 12-point improvement in the SAST scale after 12 months of administration of leuprorelin. This result is consistent with the aforementioned studies; GnRH depots are thought to act directly on the central nervous system (CNS) to suppress abnormal sexual behavior. 

There were also various hormonal side effects caused by the action on the CNS. GnRH depot leads to hypogonadism by continuously decreasing the secretion of androgen, which could result in impotence, especially in adult males, and could also reduce testicular volume and body hair [10]. Also, the long-term administration of GnRH depot leads to a decrease in bone density [10]. Besides this, the long-term administration of GnRH depot can lead to various physical side effects, such as calcium loss, hypertension, hepatotoxicity, and weight gain. Nevertheless, GnRH depot is recommended as one of the most effective therapeutic methods for paraphilia [13]. Most side effects from GnRH agents have been reported to be reversible when the drug administration is stopped. In this case, mild weight gain, appetite enhancement, and a subjective feminine appearance change were observed due to drug side-effects, but there were no other side-effects.

Bone loss was most severe at 12 months after the administration of ADT and then more slowly decreased in the long-term ADT treatment [13,14]. A retrospective population-based study showed that ADT was associated with 19.4% of fracture risk, as compared with 12.6% in those not receiving ADT [15,16]. The management of bone health should include appropriate calcium/vitamin D intake, regular weight-bearing exercise, stopping smoking, and limiting alcohol consumption to <2 standard drinks per day [17]. Bisphosphonates, including pamidronate, alendronate, risedronate, and zoledronic acid, prevented ADT induced BMD loss in randomized controlled studies [18,19,20]. In addition, denosumab and the selective estrogen receptor modulator toremifene decreased the incidence of fracture in men with nonmetastatic prostate cancer undergoing ADT treatment [21,22]. In this case, we maintained 70 mg alendronate/per week and calcium/vitamin D at 500–1200 mg/800–1400 IU/per day. Despite GnRH administration for a paraphilia with osteoporosis, there was no evidence of aggravation of osteoporosis in this case. There were many causes of osteoporosis in this patient, but the main cause was thought to be hyperthyroidism. In previous studies, hyperthyroidism has also been reported to affect GnRH in animal studies but has not been confirmed in human studies [23]. On the other hand, GnRH caused hypogonadism, which can exacerbate osteoporosis [7]. He had no fracture during this treatment period. Luteinizing hormone (LH) and follicle-stimulating hormone (FSH) were increased above the reference level at the beginning of treatment, which seems to be due to aging. Also, there was a temporary rise in PTH, but this phenomenon appeared to be secondary to changes in subclinical hyperthyroidism and vitamin D change and then soon stabilized.

In this case, we found the improvement of paraphilic symptoms after the administration of the GnRH agent, and it was confirmed that no significant side-effects appeared in the osteoporosis and thyroid disease. However, the results are difficult to generalize. This was only a case report for one year, which is not a sufficient period. Therefore, future studies will require a sufficient number of years. In the future, GnRH studies will require the research plan to be elaborately designed and use accurate diagnostic criteria and a large scale. In addition to studies on GnRH agent efficacy, various safety studies on endocrinological and mental side effects are also necessary. Because osteoporosis and thyroid disease are clinically common diseases, the presence or absence of GnRH side-effects for these diseases has a very important clinical implication. GnRH agent is currently used in sexual offenders with paraphilia, but its widespread use has been limited because of its various side effects. We hope that GnRH will be widely used as an alternative/complementary therapy in patients with paraphilia with these osteoporosis and thyroid diseases.

## Figures and Tables

**Table 1 medicina-55-00705-t001:** Laboratory findings and medications of a sexual offender with paraphilia.

Reference	Baseline	6 Months	12 Months
Free T4 (0.78–1.94 ng/dL)	0.96	1.13	0.92
TSH (0.25–4 mIU/L)	0.02	1.48	4.23
Anti-TSH-R (0–1.5 IU/mL)	3.9	2.0	0.3
Anti-TPO (0–100 IU/mL)	139.9	-	-
Anti-Tg (0–70)	322.8	-	-
LH (0.5–10 IU/L)	17.75	0.10	0.20
FSH (1.3–11.5 IU/L)	32.02	9.93	11.86
Testosterone (1.34–6.25 ng/mL)	3.13	0.11	0.23
Prolactin (2.1–17.7 ng/mL)	16.84	-	-
25(OH)D (ng/mL)	23.6	27.5	11.0
PTH (10–57 pg/mL)	63.3	12.9	-
Serum C-telopeptide (<0.704 ng/mL)	0.033	0.084	0.155
Hemoglobin (13–17 g/dL)	13.9	11.8	11.4
**BMD** [g/cm^2^/T-score, (percent change from baseline)]			
Lumbar spine 1–4	0.704/−2.7	0.671/−2.9 (−4.7%)	0.704/−2.7 (0%)
Femur neck	0.531/−2.5	0.521/−2.6 (−1.9%)	0.529/−2.6 (−0.4%)
Total femur	0.717/−1.6	0.673/−2.0 (−6.1%)	0.750/−1.4 (+4.6%)
Random glucose (55–115 mg/dL)	157	136	97
HbA1c (4–6%)	5.4		
Cholesterol (120–239 mg/dL)	166	174	196
Triglyceride (1–200 mg/dL)	159		
HDL-Cholesterol (35–55 mg/dL)	49		
LDL-Cholesterol (<130 mg/dL)	105		
AST (4–37 U/L)	22	21	21
ALT (4–41 U/L)	24	16	14
Alkaline phosphatase	81	80	72
(40–129 IU/L)			
Total bilirubin (0.2–1.2 mg/dL)	0.56	0.28	0.16
Height (cm)	164	164	164
Weight (kg)	65	65	65
**Medications**			
Alendronate (mg/week)	70 mg/Wk	70 mg/Wk	70 mg/Wk
Calcium (mg/day)	1200 mg/D	500 mg/D	500 mg/D
Vitamin D (mg/day)	800 IU/D	1000 IU/D	1400 IU/D
propylthiouracil (mg/day)	100 mg/D	-	-
Methimazole (mg/day)	10 mg/D	7.5 mg/D	2.5 mg/D
Leuprorelin (mg/month)	3.75 mg/Mo	3.75 mg/Mo	3.75 mg/Mo

Free T4 = free thyroxine, TSH = thyrotropin, anti- TSH-R = anti TSH receptor antibody, anti-TPO = anti thyroid peroxidase antibody, anti-Tg = anti thyroglobulin antibody, LH = luteinizing hormone, FSH = follicle stimulating hormone, 25(OH) D = 25 hydroxy vitamin D, PTH = parathyroid hormone, HbA1c = glycated hemoglobin, BMD = bone mineral density.

**Table 2 medicina-55-00705-t002:** Psychiatric finding of a sexual offender with paraphilia.

	Baseline	6 Months	12 Months
CGI-S	5	2	1
CGI-I		1	3
SAST	13	2	1
WSFQ	13	2	1

CGI-S = clinical global impression severity, CGI-I = clinical global impression improvement, SAST = sex addiction screening test, WSFQ = Wilson sex fantasy questionnaire.

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
