# Peer review of "A Case of Paraphilia with Osteoporosis and Administered with Depot Leuprorelin"

_medicina, 2019, doi:10.3390/medicina55100705_

Round 1
Reviewer 1 Report
The author should focus on the discussion part in terms of writing. They should rewrite the section again to achieve the interest for the readers The author really needs to work on the english writing with a help of native english speaking person or someone else. That will flurish the case report which they have submitted.Author Response
Response to Reviewer 1
English language and style
(x) Extensive editing of English language and style required
Response: If the manuscript is minor revision or acceptance, we will be glad to send it to the English revision agency recommended by MDPI as soon as possible.
|
Yes |
Can be improved |
Must be improved |
Not applicable |
|
|
|
|
|
|
|
|
Is the research design appropriate? |
( ) |
( ) |
(x) |
( ) |
|
|
|
|
|
|
|
|
|
|
|
|
|
Are the conclusions supported by the results? |
( ) |
( ) |
(x) |
( ) |
Is the research design appropriate?
Response: This study was case study. Although this treatment started by a court order of the Ministry of Justice, we gave his informed consent for inclusion before he participated in the study. So, we had an open (not blind, not control group) study.
Are the conclusions supported by the results?
Response: Because this study was a case study, it was lacking as evidence of therapeutic implications. However, we tried to support the conclusion by various supplementing the implications with the discussion.
Comments and Suggestions for Authors
Point 1: The author should focus on the discussion part in terms of writing.
They should rewrite the section again to achieve the interest for the readers
Response 1: I am appreciate your kind comments
We revised this manuscript especially on discussion section as your comments.
Point 2: The author really needs to work on the English writing with a help of native english speaking person or someone else. That will flurish the case report which they have submitted.
Response 2: Thank you for your kind comments.
This manuscript was first modified by an English language correctional institution (Harisco co.) in Korea, but it will nevertheless be sent to a specialized institution for editing in English. However, if the manuscript is minor revision or acceptance, we will be glad to send it to the English revision agency recommended by MDPI as soon as possible.
Reviewer 2 Report
This is an interesting case report, especially since it is interdisciplinary.
However, there are some uncertainties in what it wants to tell us. I don´t think that the diagnosis “hyperparathyreoidism” is correct/valid in the current setting (see below).
Further, there are some language problems (both minor typing errors as well as real linguistic inaccuracies) that need to be addressed.
For example, I don´t understand the following sentence: „…that has been difficult to remission like habit and impulse control disorder.“ (Line 15-16)
What do you mean by „selective therapeutic method“? (Line 16-17)
Please introduce/explain abbreviations, for example „DSM-5“ (line 31). I am an endocrinologist and not familiar with psychological questionnaires or diagnostic scores.
Line 39-48: A detailed description of the legal situation of paraphilia is not necessary in this context. Please consider to omit. Please rather describe the therapeutic goal instead, and whether GnRH analogues are a licenced therapeutic option or an individual healing attempt in your case. Please describe why you chose leuprorelin. Is it a standard therapy for the disorder?
Line 67-68: You state that there are no other reports from Korea, but what about the rest of the world?
Case report:
What was the specific treatment the patient had received before?
Describing the case, please use past form, not present.
Table 1:
Why were LH and FSH increased at baseline? As a consequence of previous treatment?
Why did PTH decrease? PTH was only slightly increased in the beginning. I don´t think you can call this hyperparathyreoidism, especially since it disappears later. What form of hyperparathyreoidism?
Please include glucose and lipid levels and liver parameters. In Table 1
Line 136: Did the patient have a fracture before?
Line 143: What do you mean with “etc”. Did he have gynecomastia?
Line 144-145: „There was an irritation, which was observed in the early stage of the outpatient treatment and the treatment compliance was also appropriately maintained.“ What does that mean?
Line 147: „All subjects …“ Who is that?
Line 148: Was this a clinical study?
Discussion: Difference between triptorelin and leuprorelin?
Author Response
Open Review
English language and style
(x) Extensive editing of English language and style required
|
Yes |
Can be improved |
Must be improved |
Not applicable |
|
|
|
|
|
|
|
|
Is the research design appropriate? |
( ) |
( ) |
( ) |
(x) |
|
|
|
|
|
|
|
|
|
|
|
|
|
Are the conclusions supported by the results? |
( ) |
( ) |
(x) |
( ) |
Is the research design appropriate?
Response: This study was case study. Although this treatment started by a court order of the Ministry of Justice, we gave his informed consent for inclusion before he participated in the study. So, we had an open (not blind, not control group) study.
Are the conclusions supported by the results?
Response: Because this study was a case study, it was lacking as evidence of therapeutic implications. However, we tried to support the conclusion by various supplementing the implications with the discussion.
Comments and Suggestions for Authors
This is an interesting case report, especially since it is interdisciplinary.
However, there are some uncertainties in what it wants to tell us.
Point 1: I don´t think that the diagnosis “hyperparathyreoidism” is correct/valid in the current setting (see below).
Response: We did not think that this case was “hyperparathyroidism”, this case was Grave’s disease (hyperthyroidism).
Point 2: Further, there are some language problems (both minor typing errors as well as real linguistic inaccuracies) that need to be addressed.
For example, I don´t understand the following sentence: „…that has been difficult to remission like habit and impulse control disorder. (Line 15-16)
Response: Thank you for your kind comments. This manuscript was first modified by an English language correctional institution (Harisco co.) in Korea, but it will nevertheless be sent to a specialized institution for editing in English. However, if the manuscript is minor revision or acceptance, we will be glad to send it to the English revision agency recommended by MDPI as soon as possible.
“ like habit and control disorder” was considered unnecessary in the sentence and deleted.
Paraphilia is a complex psychological and psychiatric disorder that has been difficult to remission like habit and impulse control disorder.
Point 3: What do you mean by “selective therapeutic method ?” (Line 16-17)
Response: we changed “ selective” to “one of the “. As an abstract, not enough to describe “selective therapeutic methods”. So several other methods are described in the ‘introduction section’. Also, the patient did not receive any special medical or psychological treatment other than grave disease as follows.
The patient was diagnosed with Graves’ disease 10 years ago, and has been taking 75-100mg/day of propylthiouracil (PTU). In addition, the patient was diagnosed with osteoporosis 10 years ago, and has been taking 70 mg of alendronate weekly and calcium 600mg/vitamin D 400 IU twice a day. According to the past history, there was no other disease relevant to internal medicine or neurology.
Point 4: Please introduce/explain abbreviations, for example “DSM-5” (line 31). I am an endocrinologist and not familiar with psychological questionnaires or diagnostic scores.
Response: Thank you for your detailed comments. We described the full name for the abbreviation as follows.
Diagnostic and Statistical Manual of Mental Disorders, fifth edition (DSM-5)
Point 5: Line 39-48: A detailed description of the legal situation of paraphilia is not necessary in this context. Please consider to omit.
Response: We deleted the “legal situation” as your comment.
Point 6: Please rather describe the therapeutic goal instead, and whether GnRH analogues are a licenced therapeutic option or an individual healing attempt in your case.
Response: We followed the guidelines of the World Federation of Societies of Biological psychiatry (WFSBP). Although legally mandatory treatment was required, the choice of treatment option was determined by the therapist. So we added these sentences as follows. “According to the guidelines of the World Federation of Societies of Biological psychiatry (WFSBP), we rated as Level 5 because of his high risk of sexual assault crime and high degree of paraphilic trait after release. We evaluated the patient's state more seriously than Stage 4, which is the general sex hormone oral administration stage, and administered GnRH injection according to the guideline.”
Point 7: Please describe why you chose leuprorelin. Is it a standard therapy for the disorder?
Response: Leuprorelin/triptorelin/goserelin are same type of GNRH. Therefore, there is no difference in mechanism of action. We will add it to the text if the reviewer wants additional explanation. “We chose luprolelin because it is the most widely used medicine. It is also highly reliable because of its European certification in the treatment of precocious puberty.”
Point 8: Line 67-68: You state that there are no other reports from Korea, but what about the rest of the world?
Response:
To the best of our knowledge, this is the first time that GnRH injection has been used for paraphilia with osteoporosis/hyperthyroidism.
Point 9: Case report:
What was the specific treatment the patient had received before?
Response:
This patient had no specific treatment for other diseases or paraphilia other than general hyperthyroidism and osteoporosis.
The patient was diagnosed with Graves’ disease 10 years ago, and has been taking 75-100mg/day of propylthiouracil (PTU). In addition, the patient was diagnosed with osteoporosis 10 years ago, and has been taking 70 mg of alendronate weekly and calcium 600mg/vitamin D 400 IU twice a day. According to the past history, there was no other disease relevant to internal medicine or neurology.
Point 10: Describing the case, please use past form, not present.
Response: Thank you. We revised the sentence into the past tense form
Point 11: Table 1:
Why were LH and FSH increased at baseline? As a consequence of previous treatment?
Response:
We think that Leutenizing hormone (LH) and follicle-stimulating hormone (FSH) were increased above the reference level due to aging.
Point 12: Why did PTH decrease? PTH was only slightly increased in the beginning. I don´t think you can call this hyperparathyreoidism, especially since it disappears later. What form of hyperparathyreoidism?
Please include glucose and lipid levels and liver parameters. In Table 1
Response: The patient does not have parathyroid disease.
There was a rise in PTH in the table 1, but this was a secondary rise above thyroid and vitamin D primarily. After both problems were corrected, PTH declined to normal.
An initial serum parthyroid hormone (PTH) level was slightly increased secondarily by subclinical hyperthyroidism state and low vitamin D level, but decreased within the reference range after these problems were resolved.
Point 13: Line 136: Did the patient have a fracture before?
Response: The patient never had a fracture after the GnRH injection and medical treatment.
Point 14: Line 143: What do you mean with “etc”. Did he have gynecomastia?
Response: The patient had feminine appearance changes (facial contour changes, smooth skin) but no gynecomastia was observed. Therefore, etc is deleted from the sentence.
the patient exhibited a rather feminine appearance change, such as skin smooth and soft, facial contour turning round. etc.).
Point 15: Line 144-145: “There was an irritation, which was observed in the early stage of the outpatient treatment and the treatment compliance was also appropriately maintained.” What does that mean?
Response: We revised “an irritation” to “an emotional irritation”. Sorry for the confusion in terms.
Point 16: Line 147: „All subjects …“ Who is that?
Sorry for the wrong sentence. We revised as follow.
A patient gave their informed consent for inclusion before he participated in the study.
Point 17: Line 148: Was this a clinical study?
Response: Yes. This study was a clinical case study.
Point 18: Discussion: Difference between triptorelin and leuprorelin?
Response: Both drugs are a type of GNRH. Therefore, there is no difference in mechanism of action. We will add it to the text if the reviewer wants additional explanation. “We chose luprolelin because it is the most widely used medicine. It is also highly reliable because of its European certification in the treatment of precocious puberty.”
The representative commercial drugs of GnRH analogues are goserelin (3.75 mg and 11.25 mg), triptorelin (3.75 mg and 11.25 mg), which was recently approved in Europe for the treatment of male patients with severe sexual abnormality, and leuprorelin.
Among them, leuprorelin (leuprolide acetate 3.75 mg and 11.25 mg), which is one of the luteinizing hormon releasing hormone (LHRH) analogue depot agents, has also been used as a selective treatment for central precocious puberty.
Round 2
Reviewer 1 Report
Thanks for revised writing of the discussion part. But I still feel the english correction for the whole manuscript is needed.
Author Response
Review Report Form 1
English language and style
(x) Extensive editing of English language and style required
Comments and Suggestions for Authors
Point: Thanks for revised writing of the discussion part. But I still feel the english correction for the whole manuscript is needed.
Response: Also, we just sent this manuscript to the MDPI English revision agency (English Editing Articles Status ID 12993) and are waiting for the revision. Thank you for your kind comments.
Point: Does the introduction provide sufficient background and include all relevant references?
Response: We revised the references, deleted the sentences about legal situation because to lightening of clinical context.
Point: Are the methods adequately described?
Response: We further described that our treatment followed the guidelines of the guidelines of the World Federation of Societies of Biological psychiatry (WFSBP), .
Point: Are the results clearly presented?
Response: We further described and supplemented the lab data such as cholesterol, Liver function data in the table. Also, we added the reason for choosing Leuprorelin among the GnRHs,
Point: conclusions supported by the results?
Response: We emphasized the clinical implications of this case in the discussion area. And we added the reason why the PTH initially increased in the case, and the reason why the LH/FSH rose. Among the side effects, we added the progress of fracture was described, and revised the main causes of osteoporosis, thyroid disease may affect osteoporosis, and thyroid hormone may affect GnRH etc

Reviewer 2 Report
Dear authors,
thank you for your answers and revising the manuscript.
I´m sorry to have confused hyperthyroidism and hyperparathyroidism. I suggest to not write „osteoporosis/hyperthyroidism“ together, since other people might also get confused like me. Rather write „osteoporosis and hyperthyroidism“.
What is the most likely cause of osteoporosis in your case? Do you think that hyperthyreoidism is the main reason?
Is there any evidence for an influence of the thyroid by GnRH-analoga? If not, I would suggest to omit hyperthyroidism from the title.
Line 46-49: I do not understand these two sentences. Maybe they may also be left out, since they do not seem tob e directly associated to the case?
These is still some confusion with past and present forms. Table 1 shows … is present, since the table shows something. In contrast, the described patient had this ot that value or treatment or whatever. In the introduction and discussion, mostly the present form should be used, since for example leuprorelin is still a therapeutic option …
Author Response
Review Report Form 2
Point: Is the research design appropriate?
Response: This patient was in a legal situation of court order and parole/probation, but regardless of this situation, we made a therapeutic decision using the World Federation of Societies of Biological psychiatry (WFSBP)’s guidelines and with the patient's written consent. The study was conducted in accordance with the Declaration of Helsinki, and the protocol was approved by the Ethics Committee of Dankook University Hospital (Project identification code: DKU 2015-12-014). This study is an open case study.
Point: thank you for your answers and revising the manuscript.
I´m sorry to have confused hyperthyroidism and hyperparathyroidism. I suggest to not write „osteoporosis/hyperthyroidism“ together, since other people might also get confused like me. Rather write „osteoporosis and hyperthyroidism“.
Response: Sorry for the confusion. I thought we had a good reason for misunderstanding, and we changed the title like “osteoporosis and hyperthyroidism”. Thank you.
Point: What is the most likely cause of osteoporosis in your case? Do you think that hyperthyreoidism is the main reason?
Is there any evidence for an influence of the thyroid by GnRH-analoga? If not, I would suggest to omit hyperthyroidism from the title.
Response: Hyperthyroidism is a major cause of osteoporosis and is considered to be the most important cause in this case.
Nevertheless, the results of whether thyroid disease affects GnRH have been studied only in animal studies and no human studies. Therefore, as your recommendation, we decided to exclude hyperthyroidism from the title. Thank you.
We added the following revised descriptions to the discussion.
“There were many causes of osteoporosis in this patient, but the main cause was thought to be hyperthyroidism. In previous studies, hyperthyroidism has also been reported to affect GnRH in animal studies but has not been confirmed in human studies. On the other hand, GnRH caused hypogonadism, which can exacerbate osteoporosis [7].”
Point: Line 46-49: I do not understand these two sentences. Maybe they may also be left out, since they do not seem to be directly associated to the case?
Response: I decided to rule out the legal situation, but this remained. Regardless of the legal situation, it was clinically guided treatment and with patient consent. So we deleted this sentence. So we deleted these sentences.
The pharmacologic treatment is mandatory, which is decided by the request of prosecutor and the order of the court. However, they were detained at the National Forensic Hospital and after they leave, they were under the monitor of parole or probation [4].
Point: These is still some confusion with past and present forms. Table 1 shows … is present, since the table shows something. In contrast, the described patient had this ot that value or treatment or whatever. In the introduction and discussion, mostly the present form should be used, since for example leuprorelin is still a therapeutic option …
Response: We read the manuscript once again and changed the tense to the past tense. Thank you for your kind opinion.
Also, we just sent this manuscript to the MDPI English revision agency (English Editing Articles ID 12993) and received it and revised for this final manuscript.
